# Wearable Fiber Optic Technology Based on Smart Textile: A Review

**DOI:** 10.3390/ma12203311

**Published:** 2019-10-11

**Authors:** Zidan Gong, Ziyang Xiang, Xia OuYang, Jun Zhang, Newman Lau, Jie Zhou, Chi Chiu Chan

**Affiliations:** 1Sino-German College of Intelligent Manufacturing, Shenzhen Technology University, Shenzhen 518118, China; xiangziyang94@163.com (Z.X.); chenzhichao@sztu.edu.cn (C.C.C.); 2Department of Electrical Engineering, The Hong Kong Polytechnic University, Hong Kong 999077, China; xia.ouyang@connect.polyu.hk; 3School of Design, The Hong Kong Polytechnic University, Hong Kong 999077, China; alice.zj.zhang@connect.polyu.hk (J.Z.); newman.lau@polyu.edu.hk (N.L.); 4Apparel & Art Design College, Xi’an Polytechnic University, Xi’an 710048, China; xianzj99@163.com

**Keywords:** fiber optic technology, smart textile, textile fabrication techniques, wearables

## Abstract

Emerging smart textiles have enriched a variety of wearable technologies, including fiber optic technology. Optic fibers are widely applied in communication, sensing, and healthcare, and smart textiles enable fiber optic technology to be worn close to soft and curved human body parts for personalized functions. This review briefly introduces wearable fiber optic applications with various functions, including fashion and esthetics, vital signal monitoring, and disease treatment. The main working principles of side emission, wavelength modulation, and intensity modulation are summarized. In addition, textile fabrication techniques, including weaving and knitting, are discussed and illustrated as combination methods of embedding fiber optic technology into textile fabric. In conclusion, the combination of optical fibers and textiles has drawn considerable interest and developed rapidly. This work provides an overview of textile-based wearable fiber optic technology and discusses potential textile fabrication techniques for further improvement of wearable fiber optic applications.

## 1. Introduction

Textiles and fibers have been developing over thousands of years with the original function of keeping people warm. Subsequently, people began pursuing fashion and esthetics. Consequently, each historical period has its own textile and apparel characteristics. A new generation of smart textiles has recently emerged along with progress in science, technology, and interdisciplinary fields; these smart textiles enable smart systems to be directly worn on the soft and curved human body [1,2]. With advancements in electronic science and technology in the recent decades, several microelectronic devices have been integrated with textiles for wearable applications; which are equipped with the functions of monitoring, communication, therapy, assistance, and entertainment, among others [3,4,5,6,7,8]. Continuous explorations in this technical field have created a new market for the wearable technology industry. Many opportunities have also been created, attracting thousands of research institutions and enterprises to develop new technologies and products, including fiber optic technology-based smart textiles. 

Optical fibers were already known in the 1960s for light transmission but not for signal transmission. In 1966, in the UK, Kao and Hockham used optical fibers for communication in a study that could be a breakthrough in this field. Over time, fiber optic technology has evolved and been gradually applied into textiles in a wearable modality for various implementations, such as communication, display, sensing, and monitoring [9,10,11]. The fibrous appearance of optical fibers is an advantage over many other devices for application in smart textiles. Optical fibers are quite similar to traditional textile fibers; the former could be ideally processed similarly as standard textile yarns for fabric fabrication, which is the first step to generating smart textile material [12,13]. Optical fibers, especially polymer optical fibers (POFs), are flexible, small-sized, lightweight, durable, cost-effective, and immune to electromagnetic interference. Moreover, fiber optic devices are easily handled with simple connections and have good biocompatibility with the human body. Light transmission or emission could be achieved on any part of a wearable device by integrating optical fibers into suitable textile structures; that is, the functions of fiber optic technology could reach any part of the human body via a wearable modality [1,12,14,15]. These features make optical fibers an ideal material to be embedded into textile structural composites.

Today, fiber optic technology not only be exploited for light and signal transmission but also be widely utilized in sensors for good metrological properties [16]. Textile techniques enable such sensors to be wearable, while simultaneously monitoring physiological parameters (e.g., heartbeat and respiratory rate) by measuring mechanical variables [15,17,18,19,20,21,22]. This work aims to briefly introduce wearable fiber optic applications with various functions, present the main working principles, and discuss certain textile techniques that enable optical fibers to be wearable. 

## 2. Wearable Fiber Optic Technology for New Fashion 

In recent decades, the use of optical fibers has increasingly expanded to wearable technologies with a wide range of functions. Driven by the demand from young and fashion-conscious consumers for unique apparel, optical fibers have been successfully embedded into textiles for illumination [23]. Unlike traditional optical fibers for signal transmission (light is reflected inside the core, and light is emitted at the end of the fiber) (Figure 1a), optical fibers processed for wearable illuminating apparel have micro perforations on the lateral side passing through the cladding to the core, as presented in Figure 1b. Light leakage then occurs because light scatters at the perforations, thereby enabling light emission on the fiber surface. [9] The alternative is to achieve the light emission by macro-bending of the optical fiber as shown in Figure 1c where the light propagation angle α is more than critical angle $\alpha_{C}$ thus to emit part of the light out of the fiber [23]. 

Numerous wearable products with attractive and changing colors have been developed by applying this lateral light emission principle of optical fibers. A flexible fiber optic display technology was developed by Koncar (Figure 2a) on a two-dimensional surface by applying the textile weaving technique [9]. Therefore, an X-Y network of optical fibers could be achieved in a woven structure acting as a fabric display screen. The display matrix was designed such that different light sources are connected by optical fibers embedded in each designed surface unit. This developed fabric display may have great potential not only in fashion but also in information display, communication, and entertainment. In addition, on the basis of the same working principle, Shenzhen Fashion Luminous Clothing Co., Ltd. (China) has commercialized this fiber optic illuminating technology by fabricating a series of wearable clothing (e.g., jackets, dresses, and underwear) and accessories (e.g., masks, hats, ties, and bags), as presented in Figure 2b [24]. Jacquard manufacturer Se-yang Textile (Korea) has developed a similar product that can display changing patterns with variable motions and illumination levels [25]. The colors of the designed luminous fiber optic fabrics could be changed because the original color of the emitted light at the micro perforations could be mixed with the color of added irradiated guided light or reflected ambient light to present a new color [23,26].

## 3. Wearable Fiber Optic Technology for Therapy

Light and optical techniques have already been applied in clinical practice for disease treatment, and these methods have brought profound impacts on modern medicine [27]. Light therapy is commonly used for pain relief, tendinopathy injuries, metabolic diseases, and tissue repair by adopting various wavelengths of light [28]. Several fiber optic devices have been integrated with fabric materials and applied close to the human skin with excellent optical and thermal properties in light therapy. Wavelength selection is quite important in light therapy because different light wavelengths have various penetration depths on human tissues, as presented in Figure 3; varying depths of wavelengths are associated with different therapeutic effects. Near-infrared light can influence chromatophores and enhance ATP (Adenosine triphosphate) synthesis in mitochondria to speed up wound healing and stimulate hair growth [29].

Shen et al. developed a textile-based side-emitting polymer optical fiber device by weaving POFs into fabrics to emit low-level red and near-infrared lights (600–950 nm) for collagen production in human fibroblast. This device could continually provide stable optical power density and operating temperature for 10 h without any potential hazard when in contact with human skin [30]. Additionally, blue-wavelength light (430–490 nm) is usually adopted in light therapy for neonatal jaundice. Quandt et al. designed a homogeneous luminous textile from POF for long-term neonatal jaundice treatment. This weave production has tries different arrays of the POFs in various fabric structures to develop a comfortable and breathable fabric; this fabric achieved positive treatment effects and could thus be used as a wearable phototherapy device to provide simultaneous care for newborns in home treatment [31]. Additionally, textile-based fiber optic technology could also be used for photodynamic therapy (PDT). Cochrane et al. designed a textile light diffuser (TLD) by using POF and Polyester yarns to flexibly and homogeneously irradiate light on human skin to activate drugs and offer therapeutic effects in dermatology [32]. Due to the complexities of the human anatomy, the wearable capability and tunability for different wavelengths of fiber optic technology would make it prospective in medical and healthcare fields. 

## 4. Wearable Fiber Optic Sensor for Monitoring

### 4.1. Applications for Real-time Vital Signal Detection

With the continuous improvement of living standards and the increased pursuit for healthcare, optical fiber technology has been developed into a variety of wearable monitoring applications to achieve real-time sensing and to increase medical and healthcare diagnosis accuracy. [18] Such optical fiber sensors have high biocompatibility and do not produce heat; they are also not susceptible to magnetic resonance and electrical discharges [19]. Table 1 summarizes the latest wearable fiber optic monitoring applications. The table shows that respiration is one of the most commonly monitored biological signals. Continuous respiratory activity monitoring, which is closely associated with many diseases, is an essential parameter in healthcare performance evaluation. For instance, respiration-related diseases include not only respiratory disorders, such as sleep apnea, asthma, and sudden infant death syndrome, but also cardiac and psychological diseases (e.g., heart failure and stress-related panic attacks), which are linked to irregular respiration [21,33]. Therefore, real-time human respiratory data should be collected and recorded. Several studies employed flexible sensors made of optical fibers into wearable modalities and placed them on the chest, back, and abdomen to test a series of physiological parameters, including respiratory rate, respiratory period, and inspiratory and expiratory phase durations [15,18,20,21,34]. Some multi-functional optical sensors have been developed for measuring vital biological parameters other than respiration, such as heartbeat and body temperature. For example, Koyama et al. in addition to Yang et al. adopted wearable macro bending optical sensors that detect tiny body vibrations caused by heartbeat and respiration to record both respiratory and cardiac activities [19,22]. Li et al. and Fajkus et al. developed wearable fiber Bragg grating (FBG) sensor-based devices that are to be placed on the chest to measure body temperature [16,17]. In addition, the function of real-time monitoring is not limited to the trunk of the human body; vital signal could also be sensed on other parts of the body to present different healthcare statuses. For instance, Najafi and his colleagues embedded optical fiber sensors into socks for assessing plantar pressure and temperature in clinical trials (these parameters are predisposing factors for foot ulcers in patients with diabetic peripheral neuropathy) to manage the biomechanical risk factors of diabetic foot disease [35]. Moreover, Arnaldo et al. applied polymer optical fiber sensors in foot-related wearable devices for gait event detection and joint angle measurement for assistance and rehabilitation [36]. However, despite the variety of wearable applications that have been developed for real-time vital signal detection, the integration method of fiber optic technology and textiles for wearable modalities needs further refinement. Table 1 shows that in most cases, fiber optic elements are embedded into polydimethylsiloxane (PDMS) and glued by polymeric glue or integrated onto an elastic substrate that is attached directly on a wearable device. This design may limit comfort, usability, and washability. Only a few studies integrated fiber optic technology into wearable modalities through textile techniques. In conclusion, wearable fiber optic monitoring applications play an important role in constant supervision and diagnostic decision-making due to their continuous measurement capability. Wearable monitoring devices would have great potential not only in healthcare management and rehabilitation but also in sports training (enhanced athletic performance) [37]. Additionally, more textile techniques could be adopted in the future by such applications to achieve comfort and flexibility, which will highly improve the usability and accuracy of devices.

### 4.2. Working Mechanism of Fiber Optic Sensors for Monitoring

Summarized from the working mechanisms of the latest wearable fiber optic technology applications in Table 1, the wavelength- and intensity-modulated sensors are most frequently adopted. For wavelength modulation, Bragg gratings-based fiber sensors, as shown in Figure 4, have great potential in applications of smart textiles, due to their advantages in high sensitivity, miniaturization, flexibility and electromagnetic immunity [38,39]. For Bragg gratings, when an broadband incident light enters the gratings, the light with central wavelength of $\lambda_{B}$ will be back-reflected. $\lambda_{B}$ can be written as
(1)$$\;\;\lambda_{B}=2n\Lambda$$
where *n* is the effective refractive index of the guided mode in optical fiber, and Λ represents the grating pitch. When variations of temperature or strain are applied to the gratings, the grating pitch Λ would change due to the thermal expansion or deformation of the fiber, and the refractive index n varies because of the thermo-optic and the strain-optic effect [40,41]. Accordingly, a variation of the Bragg reflection occurred, by which target physical parameters (e.g., temperature and strain) can be measured [42].

For the intensity modulation, Figure 5 shows the working mechanism of fiber optic micro bend sensor. Microbending loss is a type of light intensity loss caused by defects and small geometrical perturbations along the fiber axis, the deformation of which is in the order of micrometers [42,43]. Light propagating in the microbending optical fiber with intensity $I_{I}$ can be modulated by external load signals such as strain, pressure and acceleration resulting in a varied light intensity $I_{S}$. Therefore, an output intensity $I_{D}$ is obtained to monitor target physical parameters such as respiration rate, plantar pressure, or the heartbeat vibration, which present great potential in the biomedical field.

## 5. Textile Fabrication Techniques for Wearable Fiber Optic Technology

Several textile fabrication techniques, such as weaving, knitting, and non-weaving methods, can be used to embed fiber optic technology into textile fabric and make fiber optic technology wearable. Comfort, flexibility, usability, and accuracy of relevant devices could thus be highly improved. 

### 5.1. Weaving

Warp yarns and weft yarns are interlaced one by one in a basic woven structure, as presented in Figure 6a. In most cases, optical fibers are woven into a fabric in unbent condition or with a limited bending angle to ensure effective transmission and sensing functions [9,44,45]. Optical fibers and standard textile yarns are commonly fabricated via a handloom by interlacing in accordance with design rules (e.g., plain, twill, and sateen (Figure 6b). In addition, optical fibers in different woven structural designs would have various mechanical and sensing properties. For instance, Wang et al. reported that the side-emitting properties of optical fibers in sateen woven structures are significantly higher than those in plain and twill woven structures [46]. Moreover, optical fiber properties could also influence the finalized smart textile performance. For instance, commonly used and commercially available optical fibers have the diameter range from 250 µm to 3000 µm [47], an relatively larger diameter of optical fibers may induce high rigidity of smart textile, meanwhile a relatively smaller diameter would cause low light intensity and low shear resistance. [11] By adopting weaving techniques, Quandt and his colleagues (2017) interlaced optical fibers into fabric following a variety of weave designs (e.g., plain weave, plain weave alternating with Trevira CS, Satin 2/2(2), Satin 3/3(3), and Satin 6/6(6)) to provide wearable phototherapy on skin for Neonatal jaundice (hyperbilirubinaemia) [31]. In addition, optical fibers could also be embedded into fabric through embroidered techniques on woven substrates. As shown in Figure 6c, textile techniques of embroidering, such as soutache and schiffli, have been reported by Selm et al. (2007) and Quandt et al. (2015) to be applied in wearable fiber optic applications for the use of body-monitoring, health supervision and photodynamic therapy [37,48].

The preceding section discusses monolayer woven structures with the basic sectional view presented in Figure 7a, in which the gray dots are warp yarns, and the black lines are weft yarns. Woven fabric materials could also be fabricated into multilayer structures. Koncar (2005) integrated optical fiber with textiles by adopting a two-layer basic-velour fabric to develop a wearable fabric display. This woven structure not only help to make embedded optical fibers as visible as possible, but also keep those fibers to be sufficiently consistent with the whole fabric [9]. The distinguished multilayer structural configuration can eliminate the delamination problem of common laminated composites. Therefore, this type of fabric usually exhibits high strength, large stiffness, high impact, and damage resistance; hence, such materials are frequently applied in textile-based wearable applications and functional apparel. A two-layer woven fabric (Figure 7b) consists of two layers of warp yarns separated by one layer of weft yarns. Two sets of weft yarns interlace in the fabric up and down across the warp yarns. Layers of the woven fabric materials could be freely designed in accordance with practical use, as shown in Figure 7c. Even if the layers remained the same, the interlacing methods of weft yarns could vary, as seen in Figure 7d–f. In particular, a three-dimensional woven fabric is formed by connecting multilayer fabrics together by binding yarns, which are also called bundled yarns, and Z-direction yarns. These yarns can be further divided into warp and weft yarns according to the connection method for the different layers. The portion of the woven (latitude) yarn of the fabric is referred to as the splicing or splicing weft. On the basis of the different interlacing manners, and the inclination angle of the binder yarn, the warp layer, and the weft layer, the orthogonal structure is divided into the whole orthogonal (Figure 7c) and the interlayer orthogonal (Figure 7d), and the angular interlocking structure is divided into an integral angle interlock (Figure 7e) and an interlayer angle interlock (Figure 7f). A wide variety of orthogonal structures and angular interlocking structures are obtained by varying the numbers of layers of warp and weft yarns, and the length and distribution of the binder yarns. Multilayer joints can also be connected by warp yarns. The interlacing methods are roughly divided into two types, namely, binding yarns (Figure 7g) and warp yarn self-interlacing (Figure 7h). Yarn density in each layer could vary. For instance, in the three-layer woven structure in Figure 7g, the yarn density of the top and bottom layers is higher than that in the middle layer, thereby forming a hollow structure between the layers. Contrary to the binding yarns (Figure 7g), the warp yarn self-interlacing method have the same density among layers. Consequently, the mechanical properties are uniform. Various interlacing methods and fabric layers can be customized in design for different sensing applications.

Multilayer woven fabric structures offer wide choices for creating wearable optic fiber sensing technology. Optical fibers can be fabricated into textile products with low bending situation and reduced deformation by adopting the suitable multilayer woven structure in Figure 8. Figure 8a illustrates an optical fiber embedded into a two-layer woven structure acting as a weft yarn between two layers of warp yarns. When the number of layers increase, as presented in Figure 8b, optical fibers can be flexibly applied between any two warp layers regardless of the adopted interlacing method. Furthermore, in the integration of optical fibers into a multilayer hollow structure, fibers can be inserted into hollow paths acting as a yarn in the warp, as shown in Figure 8c. 

Various structural designs and layer configurations can provide not only added possibilities for wearable fiber optic technology but also enrich relevant applications in a user-friendly and competitive manner. In the development of a textile-based fiber optic material, textile yarns in different layers (above, below, or next to the embedded optical fibers) could be designed using different types of materials (e.g., conductive and waterproof). Thus, the developed wearable fiber optic application would be equipped with additional functions, and the usability is considerably improved. Additionally, more than one woven structure can exist in a single piece of fabric material. The woven fabric in Figure 9 was designed to cover six- and two-layer structures. Consequently, hollow paths form on the surface of the fabric. In this case, optical fibers could be embedded into those paths on the fabric surface if relevant application requires sensing fibers to be tightly close to the human body. 

### 5.2. Knitting

Knitting is another textile technique used to fabricate smart textiles. Knitted fabric is typically constructed with loops that are interconnected by courses and wales. In courses, threads go horizontally in the fabric; in wales, threads run vertically. Weft knitting is a commonly used construction for optical fiber integration; this method increases the elongation percentage of smart textile application. [48] This textile technique (Figure 10) is used to fabricate knitted fabric material where stiches run horizontally from left to right. Optical fibers in knitted structures need to bend more than those in woven structures to form loops in courses. Thus, the linear density and bending resistance of yarns in knitted structures should be kept low to protect the bending areas. When fiber optic technologies need to be incorporated into fabrics and applied on the human body, a series of physical–mechanical properties, including bending, stretch rate at break, and tenacity, must be considered and tested along with ergonomic factors [49].

Except for the weft knitting technique, where optical fibers act as courses, laid-in structural knitting designs would be suitable for wearable fiber optic material fabrication, where optical fibers will not bend considerably. Figure 3 shows laid-in knitting structural designs, with the optical fibers marked yellow. Figure 11a presents the single jersey hopsack structure; a 1 × 1 optical fiber inlay is located between each plain ground course [50]. The inlay rules and positions of optical fibers could be flexibly arranged on a plain weft knitting structure, as shown in Figure 11b,c, or the inlay direction can be changed to warp, as seen in Figure 11d, to lay optical fibers between the plain ground wales. 

The laid-in structure appears not only in weft-knitted fabric materials; such structure could also be applied in warp-knitted material. Figure 12a shows the warp knitted structure where optical fiber could be horizontally or vertically integrated into by using the laying-in techniques, as illustrated in Figure 12b,c. As warp knitting is a knitting method in which yarns go through the fabric lengthwise in a zigzag rule, thus, optical fibers are usually not recommended to act as interlacing yarns due to the sharp bending at interlacing point. However, optical fibers sometimes were designed to achieve side-emitting by macro-bending configuration as mentioned in Section 2, in this case the warp-knitting structure following zigzag rules with controlled sharp bending could be applied. 

## 6. Conclusions

This article conducted a comprehensive review of wearable fiber optic applications. The functions of these devices could be mainly divided into three categories: 1) fashion and esthetic purposes (to make wearables stylish), 2) disease treatment (wearable fiber optic light therapy can treat neonatal jaundice, speed up wound healing, and stimulate hair growth), and 3) healthcare monitoring (to detect real-time vital signals, such as respiratory rate, plantar pressure, and heartbeat vibration). The working mechanisms of fiber optic technology, including side emission and wavelength and intensity modulation, were briefly introduced to understand the operating principles of different wearable fiber optic applications. Various textile fabrication techniques were illustrated, as combination methods for optical fibers and textiles. In conclusion, smart textiles enable fiber optic technology to be wearable. Such fabrics highly improve the comfort, flexibility, usability, and accuracy of relevant devices and enrich them to be user-friendly and competitive.

## Figures and Tables

**Figure 1 materials-12-03311-f001:**
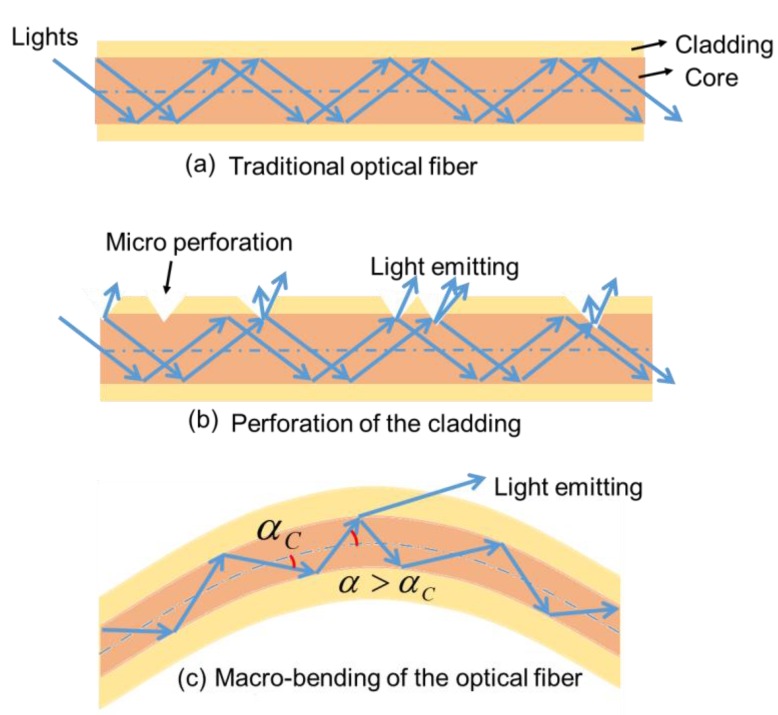
Principle of lateral light emission: (**a**) traditional optical fiber; (**b**) perforation of the cladding; (**c**) macro-bending of the optical fiber.

**Figure 2 materials-12-03311-f002:**
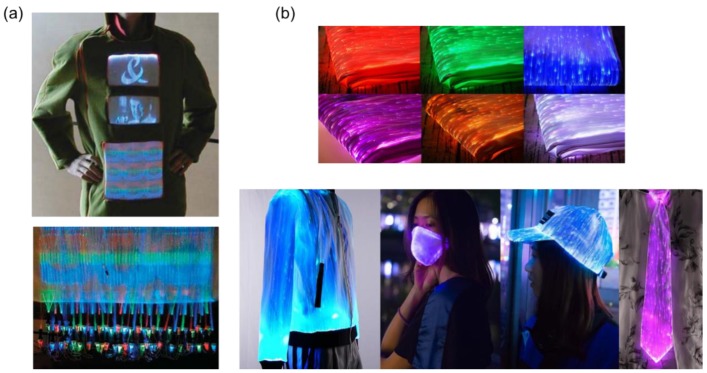
Wearable fiber optic illuminating technology: (**a**) flexible fabric display; [9] (**b**) luminous fiber optic fabrics and sample apparel. [24].

**Figure 3 materials-12-03311-f003:**
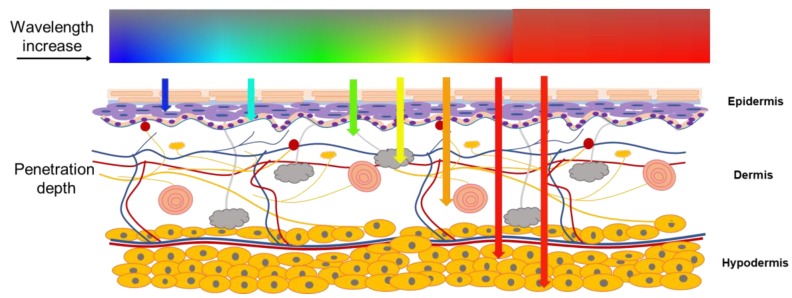
Varying depths of light penetration on tissue.

**Figure 4 materials-12-03311-f004:**
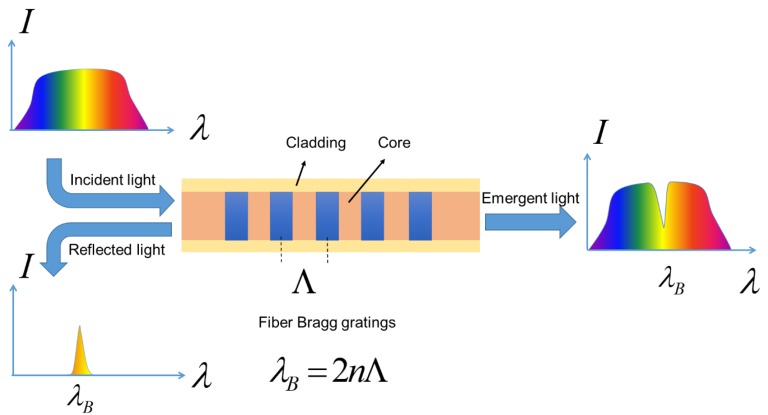
Schematic of Bragg gratings.

**Figure 5 materials-12-03311-f005:**
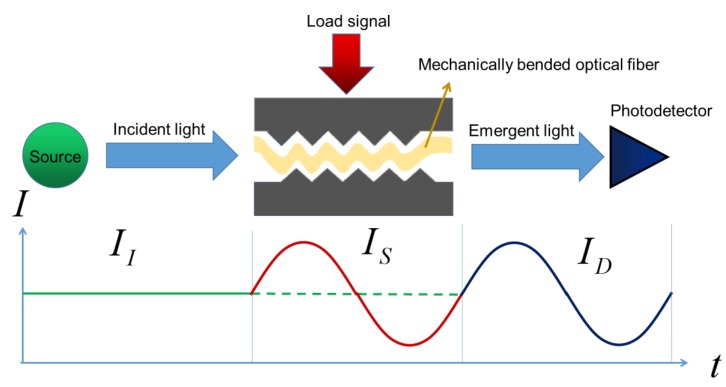
Schematic of micro bend sensor.

**Figure 6 materials-12-03311-f006:**
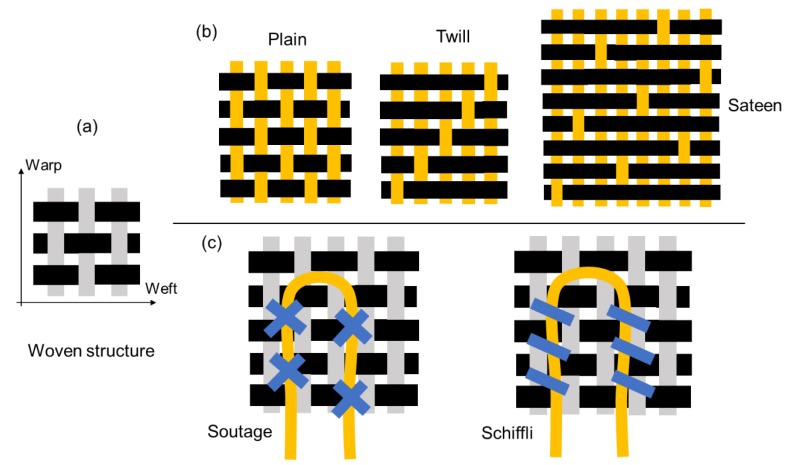
Optical fibers in woven structures: (**a**) basic woven structure; (**b**) optical fibers in plain, twill, and sateen structures; (**c**) optical fibers in woven structure with embroidering techniques (optical fibers marked yellow).

**Figure 7 materials-12-03311-f007:**
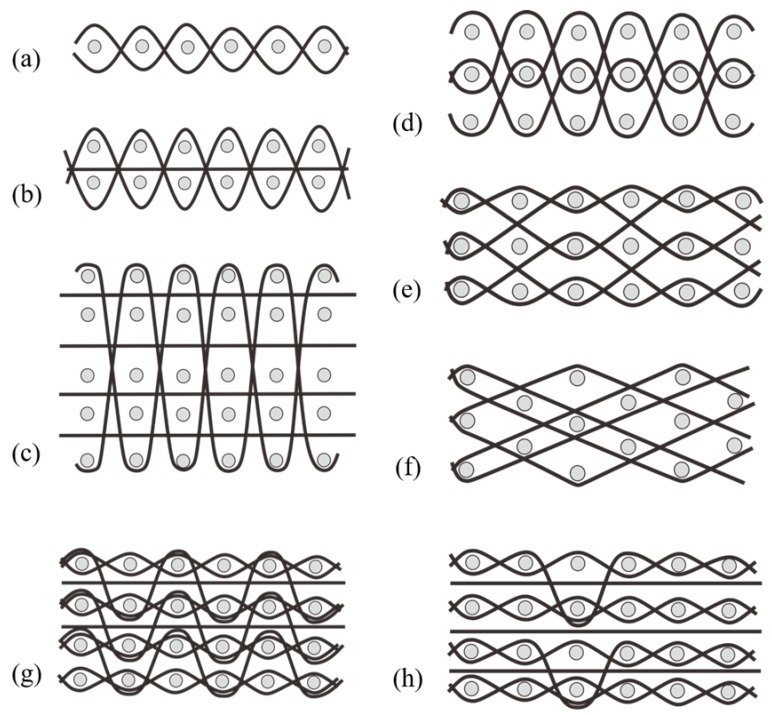
Sectional view of monolayer and multilayer wove structures: (**a**) monolayer woven structure; (**b**) two-layer woven structure; (**c**) five-layer whole orthogonal structure; (**d**) three-layer interlayer orthogonal structure; (**e**) three-layer integral angle interlock structure; (**f**) three-layer interlayer angle interlock structure; (**g**) four-layer binding structure; (**h**) four-layer self-interlacing structure.

**Figure 8 materials-12-03311-f008:**
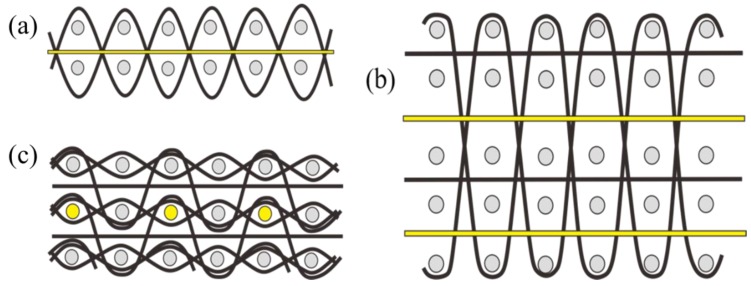
Optical fibers in multilayer woven structures: (**a**) optical fiber as a weft yarn between two layers of warp yarns; (**b**) optical fibers applied between any two warp layers in a multilayer interlacing structure; (**c**) optical fibers as warp yarns in a multilayer hollow structure.

**Figure 9 materials-12-03311-f009:**
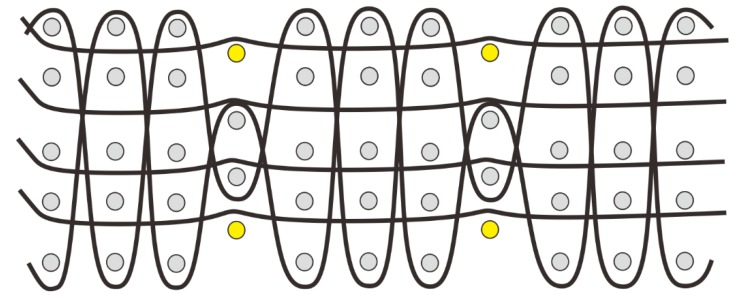
Optical fibers in woven fabric material consisting of different layer structures.

**Figure 10 materials-12-03311-f010:**
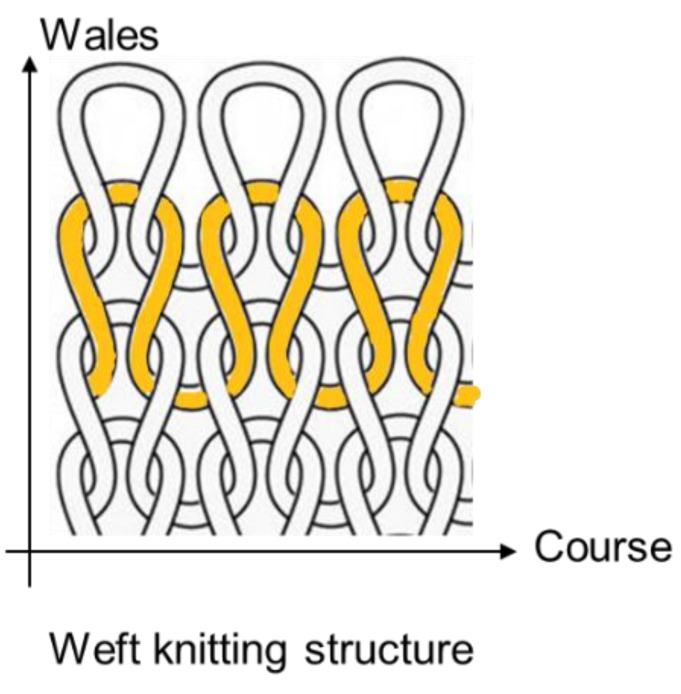
Weft knitting structure.

**Figure 11 materials-12-03311-f011:**
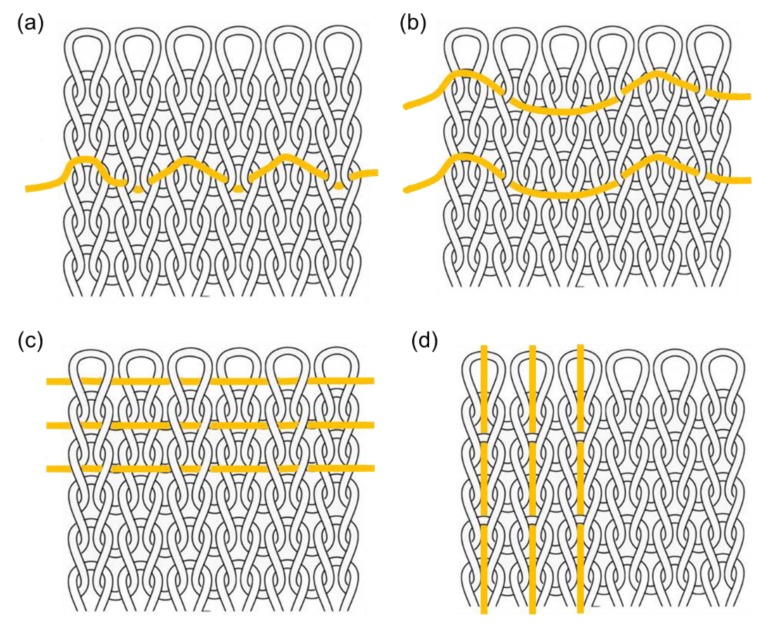
Optical fiber laid-in weft knitting designs: (**a**) single jersey hopsack structure; (**b**,**c**) other weft laid-in structures; (**d**) warp laid-in structure.

**Figure 12 materials-12-03311-f012:**
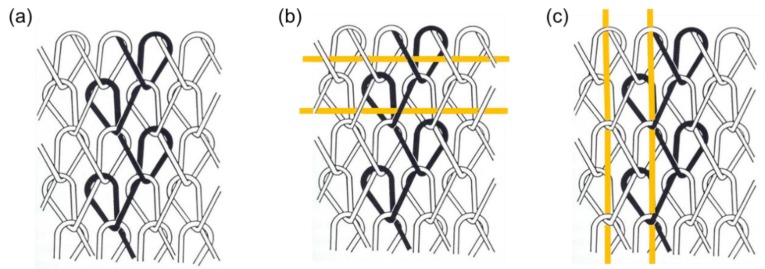
Warp-knitted structure and fiber optic application: (**a**) warp-knitted structure; (**b**) and (**c**) optical fibers laid in warp-knitted structure.

**Table 1 materials-12-03311-t001:** Wearable fiber optic technology applications in healthcare monitoring.

Reference	Working Mechanism	Application	Integration Method	Location on Body	Characteristics
Koyama et al. (2018) [22]	Intensity modulated	Heartbeat and respiration monitoring	Woven with wool fabric into garment	Chest surface	Comfort; real-time function; high accuracy; ability to sense minute-load changes
Li et al. (2018) [16]	Wavelength modulated	Wrist pulse, respiration, and finger pulse monitoring	Embedded in PDMS	Wrist, chest, and finger	High sensitivity of 0.83 kPa^−1^; real-time function; flexibility; wearability; cost-effectiveness
Arnaldo et al. (2018) [36]	Intensity modulated	Gait monitoring	Attached to insole, orthotic device, and modular exoskeleton	Foot	Flexibility; high repeatability; low cost; simple signal processing; measurement of joint angles and detection of gait events for gait assistance and rehabilitation
Lo Presti et al. (2017) [34]	Wavelength modulated	Respiratory monitoring	Glued by polymeric glue	Chest wall	Monitoring in harsh environments; ability to measure in different positions of the human body
Najafi et al. (2017) [35]	Wavelength modulated	Plantar pressure and temperature monitoring	Embedded into socks	Foot	Quick feedback; real-time function; convenience
Fajkus et al. (2017) [17]	Wavelength modulated	Body temperature, respiration, and heart rate monitoring	Encapsulated inside PDMS	Chest surface	Non-invasiveness; high accuracy; multichannel hybrid fiber optic sensor system
Hu et al. (2016) [18]	Wavelength modulated	Respiratory monitoring	Attached to seat-back	Back	Real-time function; high accuracy; low cost; convenient operation
Ciocchetti et al. (2015) [20]	Wavelength modulated	Respiratory monitoring	Glued by adhesive silicon rubber	Chest surface	Non-invasiveness; good linear response to strain; chemical inertness; small size; flexibility; MR compatibility; high accuracy in the estimation of T_R_, T_I_, and T_E_ phases and UT volumes.
Yang et al. (2015) [19]	Intensity modulated	Heartbeat respiration monitoring	Integrated onto an elastic substrate	Back or chest	Simultaneous measurement in daily activities; comfort; cost-effectiveness; high sensitivity; non-invasiveness; simple fabrication
Zheng et al. (2014) [15]	Intensity modulated	Respiration monitoring	Embedded into belt fabric	Chest or abdomen	High strain sensitivity; low hysteresis and repeatability; immunity to electromagnetic interference
Witt et al. (2012) [21]	Wavelength modulated	Respiration monitoring	Integrated into textile-based sensing harness	Abdominal and thoracic areas	Comfort; continuous measurement; testing in MR environment

FBG: fiber Bragg grating; MR: magnetic resonance; T_R:_ respiratory period; T_I_: duration of inspiratory; T_E_: expiratory; UT: upper thorax; PDMS: polydimethylsiloxane polymer

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
