# Peer review of "Wearable Fiber Optic Technology Based on Smart Textile: A Review"

_materials, 2019, doi:10.3390/ma12203311_

Round 1

Reviewer 1 Report

English language of the paper is not vetted.

The review is well constructed and explains things well.

The part concerning structures (part 5) is important but perhaps occupies too much space compared to the first three (functions).

Line 48: “micro-sized”. PMMA commercial optical fiber has a diameter range from 250 µm to 3000 µm. The word “micro-sized” is therefore controversial.

Line 48: “robust”. It depends on the solicitation. PMMA is not solvent resistant, temperature or abrasion…

Part 2: Even if for fashion, perforation of cladding (solvent or sanding), is the most common, the exceeding the limit angle is also a possibility to make flexible fabric display.

It may be well to invert part 3 and 4. Since part 2 describes the light emission (for fashion), part 3 the monitoring and part 4, again the light emission (for the medical).

Part 4: You cite phototherapy but you forget Photodynamic therapy (PDT), where light activates a drug. For instance cancer treatment in dermatology. You can find an example of textile light diffuser design for this application in this paper: C. Cochrane, S. R. Mordon, J. C. Lesage, and V. Koncar, “New design of textile light diffusers for photodynamic therapy,” Mater. Sci. Eng. C, vol. 33, no. 3, pp. 1170–1175, Apr. 2013.

You have included a scheme for FBG, modulation and lateral emission by cladding treatment. Can you add a diagram to illustrate the emission of light by exceeding the limit angle (macro bending)?

Line 248-249: Perhaps you could indicate the diameters of fiber optically used or commercially available. https://www.toray.co.jp/english/raytela/products/pro_a001.html

Although it is true that some laboratories manufacture their own optical fibers to adapt the flexibilities for example (your ref 42 and previous paper of this lab).

Line 352: “sharp bending”: this bending is sometimes desired and controlled

References

Some references are quoted twice (maybe more?). example 9 and 25, 22 and 30, 15 and 18…please check the entire bibliography!

Author Response

Please see the attachment. Many thanks.

Reviewer 2 Report

The manuscript “Smart textiles enable wearable fiber optic technology: A review” is written to be featured in a special issue : "Novel Optical Fibers, Devices and Applications" of Materials. My comments on the manuscript is the following:

The title “Smart textiles enable wearable fiber optic technology” means “Smart textiles make fiber optic technology wearable”, as the author mentioned inside the text. This is not a noun phrase, but a full sentence, which a bit unconventional for a title.

It seems to me that all figures are original, made by the authors for this manuscript, since there is no reference in any figure, except figure 2. Is this correct?

In the Introduction, the sentence “these devices are known as wearable technology” means that wearable technology is limited to textile-based devices, however, watches (iWatch), bracelets (Fitbit) and goggles (Google glasses) are also examples of wearable technology.

The subsection 3.2 is not very well written, although the part 3.1 is. In 3.2, for example, describing the incident light in Figure 3 as "λ“is not correct, λ in the figure is the label of the x-axis. Explaining the principle by a sentence “λB will vary with changes in the physical parameters targeted for testing (e.g., temperature and strain)” is maybe too simplified, and finally, the word “sensed’ might be replaced by “monitored” or “measured” in this context. The same for the word “detect” in line 193 page 8, which should be replaced by “monitored” or “measured”.

Section 5, especially from Line 258 page 11 (by the way, page number is re-counted from page 8) is a description of many fabric fabrication techniques. There are, however, not many citations in which show the state of the art of the integration of optical fibers using such techniques. In a review, readers look for citations in which they can find further information about how a particular work has been done. In this part of the review, the author discuss about how optical fibers would be integrated in textiles, without showing how exactly it has been done, when, and where. Citations 50 is a review, which raises a question if the opinion “When fiber optic technologies need to be incorporated into fabrics and applied on the human body, especially the elbows, a series of physical–mechanical properties, including bending, stretch rate at break, and tenacity, must be considered and tested along with ergonomic factors” belongs to the authors of this manuscripts, or the author of the citation 50.

There are few typos in the manuscript, such as in line 60 page 2, “functions” need to be in plural “functions”, “absorptivity” -> “absorption”. Besides, authors may consider using words like “fiber optic technology”, “fiber optic technologies”, “fiber optic sensing”, “optic fiber sensing” more consistent throughout the manuscript.

Best regards.

Author Response

Please see the attachment. Many thanks.

Round 2

Reviewer 2 Report

Dear authors,

Thank you for taking into account my comments and suggestions. In my opinion, the manuscript is now suitable for publication.

Best regards.